Subject Area:
cellular biology/developmental biology/ genetics/molecular biology/neuroscience

Keywords:
epigenetics, biomarkers, alcohol, cannabis, addiction, immune assays

Author for correspondence:
Yasminah Elsaadany Dobs
e-mail: yasmina.elsaadany@gmail.com

# The epigenetic modulation of alcohol/ethanol and cannabis exposure/co-exposure during different stages

Yasminah Elsaadany Dobs[1] and Mohamed Medhat Ali[2,3]

[1]Department of Biology and Biomedical Science, North Carolina Central University, Durham, NC, USA
[2]Biomedical Sciences Program, Zewail City of Science and Technology, Giza, Egypt
[3]Department of Medical Microbiology and Immunology, Faculty of Medicine, Mansoura University, Egypt

YED, 0000-0003-1961-7425; MMA, 0000-0002-0852-8382

Studies have reported the significant economic impact of smoking cannabis and drinking alcohol In the USA. It was estimated that the costs of cannabis-related treatment, hospitalization and loss of work-related pay have amounted to $200 billion. (Andersen AM, Dogan MV, Beach SRH, Philibert RA. 2015 *Genes* **6**, 991–1022. (doi:10.3390/genes6040991)). Data from the National Epidemiologic Survey on Alcohol and Related Conditions showed that individuals with general anxiety disorder and substance use disorder (GAD-SUD) have higher psychiatric comorbidity rates than those without substance use disorder (Alegría AA, Hasin DS, Nunes EV, Liu SM, Davies C, Grant BF, Blanco C. 2010 *J. Clin. Psychiatry* 71, 1187–1195. (doi:10.4088/JCP.09m05328gry)). Moreover, the criminal justice system is significantly impacted by this cost (Andersen AM, Dogan MV, Beach SRH, Philibert RA. 2015 *Genes* **6**, 991–1022. (doi:10.3390/genes6040991)). Despite the increasing use of cannabis, there are still too many obscure facts. One of the new areas that scientific evidence shows is impacted negatively by cannabis use is the epigenome, which is an understudied area that we are still learning about. In addition, over the past few decades, we have seen various social and healthcare changes that have raised critical questions about their ongoing roles in regulating marijuana and alcohol use. This is important because of the increasing popularity and usage across various ages especially young adults and teenagers. More than 97.5 million Americans over 12 years old have used cannabis for non-medical use despite the significant side effects, with 1 in 10 users developing cannabis dependence (Crean RD, Crane NA, Mason BJ. 2011 *J. Addict. Med.* 5, 1–8. (doi:10.1097/ADM.0b013e31820c23fa), Office of Applied Studies. 2006 Substance Abuse and Mental Health Services Administration, USA.). It was reported that 16% of substance abuse admissions in the USA were for cannabis-related symptoms, which is second only to alcohol-related disorders (Agalioti T, Lomvardas S, Parekh B, Yie J, Maniatis T, Thanos D. 2000 *Cell* 103, 667–678. (doi:10.1016/S0092-8674(00)00169-0), Soutoglou E, Talianidis I. 2002 *Science* 295, 1901–1904. (doi:10.1126/science.1068356)). Today there are thirty-one states and the District of Columbia that currently have legalized marijuana for either medical or recreational use. Data about marijuana use from NIAAA's National Epidemiologic Survey on Alcohol and Related Conditions (NESARC) indicates that 'in total, 79 000 people were interviewed on alcohol and drug use. When examined by age young adults (ages 18–21) were found to be at highest risk for marijuana use and marijuana use disorder, with use increasing from 10.5 to 21.2% and disorder increasing from 4.4 to 7.5%'. 'Given these facts, George Koob, PhD, director of NIAAA stated the importance for the scientific community to convey this information to the public about the potential hazards of marijuana and it's use'. On the other hand, according to the National Institute on Alcohol Abuse and Alcoholism, 16 million adults suffer from alcohol use disorders. To the best of our knowledge, epigenetic mechanisms have been previously studied in alcohol and cannabis abuse separately. Recent studies highlighted the molecular mechanisms that are linked

with drug-induced transcriptional regulation, behavioural abnormalities and neurodegeneration, which has emphasized the role of chromatin modification/remodelling in the generation of drug activation of certain genes and the disabling of others, and the effect of that on addiction (Maze I, Nestler EJ. 2011 *Ann. N. Y. Acad. Sci.* 1216, 99–113. (doi:10.1111/j.1749-6632.2010.05893.x); Renthal W, Nestler EJ. 2008 *Trends Mol. Med*. 14, 341–350. (doi:10.1016/j.molmed.2008.06.004)). In this review, we will give an overview of epigenome science relevant to cannabis/the endocannabinoid system and the potential of epigenetic overlap between alcohol and cannabinergic activity at different stages, to aid further investigations that could bring more treatment options to our horizon.

## 1. Introduction

The evolution of epigenetics in neuronal disorders has been a main scientific interest during the past decade [1,2]. Recent studies examining the molecular mechanisms controlling drug-induced transcriptional, behavioral and synaptic plasticity have indicated a direct role for chromatin remodelling in the regulation and stability of drug-mediated neuronal gene programmes and the subsequent regulation of addictive behaviors [3]. Intact epigenetic processes have been defined in various ways. One example from Adrian Bird in 2007 gave the general definition: 'The structural adaptation of chromosomal regions to register, signal, or perpetuate altered activity states' [4, p. 398]. This described activity dictates the readout of genetic transcription and the downstream phenotypes of organisms. Some cells are more prone to this modification than other renewable or differentiated cells. Epigenome process dysregulation caused by the effect of cannabinoids and alcohol can be observed in an individual's lifetime. The epigenome and the chromatin-related proteins are increasingly evolving in the pathophysiology of addiction and related psychiatric disorders [5]. Additionally, published scientific data have revealed a strong association of epigenome modulations that are impacted by cannabis and alcohol exposure and co-exposure. Several authors have agreed that there is a great knowledge gap to be fulfilled to advance our understanding in epigenome science to foster novel discoveries of molecules that will enable new treatment paradigms in substance abuse. Investigations over the past few decades have consolidated important mechanistic revelations about different addictions and related neuropsychiatric disorders [6,7].

## 2. Epigenetics definition and mechanism

In 1957, epigenetics was first introduced by Conrad Waddington, a biologist who studied how similar genotypes affected a wide variety of phenotypes. Later, it was revealed that the epigenome theory might affect the heritable changes in gene expression with no changes in DNA sequences [8]. Waddington concluded that epigenetic modification regulates certain cellular signalling that triggers the differentiation of stem cells into cells with identical genotypes that display different phenotypes [9]. The process of cell signalling followed by differentiation turns on varying numbers of genes resulting in various differentiated cells (e.g. cardiac muscle cells versus neuron cells).

Epigenetics could be defined as a series of biochemical processes that mediate changes in gene expressions during chromatin development with no changes in the DNA sequence [10]. There are multiple chromatins surrounding molecules that disturb gene constitution and, in turn, the downstream signalling pathways. Those molecules, if they are exposed to a certain stimulus such as drugs in a frequent manner or an addictive state, will turn some genes on or off. In turn, this exposure will trigger abnormalities in individual characteristics that are associated with different phenotypes, and behavioural abnormality is one of those phenotype changes [3,5]. These phenotype changes with no change in genotype called epigenetic changes, also known as chromatin remodelling. Epigenetics provides a mirror of the environmental experiences that are involved in our lifestyle, such as drug exposure, diet and stress. Therefore, epigenetic disturbances are linked as a marker for some abnormal biological processes [11–13]. In other words, 'epigenetics' refers to the process that modifies gene regulation with no changes in DNA codes that might result in a specific phenotype [1,2,11,12,14]. The new phenotypes have different methylation and acetylation from the normal chromatin [1,15]. One of the most vital components of the epigenome is the chromatin structure. Chromatin has been described previously as heads or beads of histone protein that the DNA is wrapped around. It is a genius way to allow an extensive amount of genetic information to be packaged in a condensed structure which also contributes to DNA stability. The basic unit of chromatin is the nucleosome, which is an octamer of histones with different groups such as H2A, H2B, H3 and H4 [16,17]. The epigenetic modifications that regulate gene expression include but are not limited to DNA methylation, nucleosome structure and positioning, posttranslational modifications of nucleosome histones, histone replacement and small RNA molecules that influence protein production, which changes the epigenetic mechanistic balance with exposure of cannabis and ethanol [18]. The addition of the methyl group to the chromatin during early transcription processes condenses the chromatin packaging to a tighter chromatin structure that later hinders the ability of other transcription factors to interact with chromatin for further transcription steps. In other words, methylation restructures the normal packaging of chromatin that would make it more condensed, while adding an acetyl group through the acetylation process is more likely to induce a relaxed chromatin structure which is more accessible to further factors during the transcriptional process [9,19]. Histone methylation is also linked with inhibition of the recruitment of other transcriptional factors that are histone residue dependent. For example, histone 3 lysine 4 (H3K4) methylation usually activates transcription while H3K9 methylation is repressive towards transcription [1,14,15]. Pandey *et al.* [15] described alcohol effects on epigenetic-mediated synaptic changes. Epigenetic markers and the enzymes that add or remove these marker change in expression throughout the development process. In studies by Kyzar *et al.* [20] there was a correlation of the epigenetic reprogramming with a lasting pathological effect in adulthood that induced an abnormal developmental process. Szutorisz & Hurd [13] described the effect of cannabis on the epigenome and molecular processes that are responsible for the selection of different cells, the tissue transcription and the associated behaviour. The author described the changes in the patterns of epigenetic markers, DNA methylation, histone modification and an individual's phenotype characteristics. DNA methylation

royalsocietypublishing.org/journal/rsob    Open Biol. 9: 180115

royalsocietypublishing.org/journal/rsob  *Open Biol.* **9**: 180115

**Table 1.** Epigenetic effects in response to cannabis exposure. From Hurd *et al.* [13].

| cannabinoid | epigenetic alteration | biological target | associated effect or consequence |
|---|---|---|---|
| cannabis | increased CpG DNA methylation at promoter | human peripheral blood cells | negative correlation between CB1R methylation and mRNA levels in schizophrenic cannabis users |
| cannabis | COMT gene genotype and promoter CpG DNA methylation | human adolescent peripheral blood cells | less likely cannabis dependence and decreased risk of psychosis |
| THC | H3K4me3, H3K9me2; promoter, gene body | adult rat brain (nucleus accumbens (NAc)) | decreased Drd2 gene mRNA levels in response to *in utero* THC exposure. |
| THC | H3K4me3, H3K9me3; promoter, gene body | adult rat brain (NAc shell) | increased Penk gene mRNA levels in response to adolescent THC exposure |
| THC | CpG DNA methylation at promoter's intergenic regions especially in gene bodies | adult rat NAc with parenteral THC exposure | altered methylation enriched in gene implicated in synaptic plasticity |
| THC | H3K4me3, H3K9me3, H3K27me3, H3K36me3; promoter, intergenic region, gene bodies. | differentiating mouse lymph node cells | genome-wide alterations in histone modifications associated with dysregulated genes and non-coding RNAs |
| THC | increased HDAC3 expression | human trophohoblast cell line BeWo | gene dysregulation during placental development |
| THC | DNA methylation at CpG islands, miRNA | cerebellum and peripheral T cells of Simian immunodeficiency virus-infected macaques. | altered DNA methylation, mRNA and miRNA expression profiles |
| THC | miRNAs | mouse myeloid-derived suppressor cells | altered mRNA, miRNA, and differentiation profile |
| THC | miRNAs | intestine of Simian immunodeficiency virus-infected macaque | altered miRNA profile and intestinal epithelial cell composition |
| exogenous anandamide | increased global DNA methylation | spontaneously immortalized human keratinocytes (HaCaT cell line) | decreased expression of differentiation-related genes and altered cell differentiation |
| exogenous anandamide | miRNAs | mouse lymph node cells | altered interleukin production and inflammatory response |
| HU-210, JWH-133 cannabinoid agonists | H3K4me3; global levels | Cb1R- and CB1R expressing human glioma stem-like cells (U87MG and U373MG lines) | induction of differentiation, inhibition of gliomagenesis |
| HU-210, cannabinoid agonists | miRNAs | adolescent rat brain (entorhinal cortex) | altered miRNA profile |

(5-methylcytosine), a covalent modification to DNA, and histone acetylation were found to occur repeatedly at specific loci. The interactions between genes and the surrounding molecules that produce certain phenotypes are one example of epigenetic interaction. Combining all these theories expands the knowledge showing how similar genotypes, with the same codons, can result in different characteristics that produce endless varieties of phenotypes [21,22]. Table 1, from Szutorisz & Hurd [13] summarizes the epigenetic modifications that regulate the endocannabinoid system via targeting its individual

components as well as downstream targets of a variety of cellular reactions.

# 3. What is the cannabinoid system?

Figure 1 shows a schematic model of the endocannabinoid system in the brain. Anandamide, also known as *N*-arachidonoylethanolamine (AEA), and 2-arachidonoylglycerol (2-AG) are synthesized in the postsynaptic membrane and then act

royalsocietypublishing.org/journal/rsob Open Biol. **9**: 180115

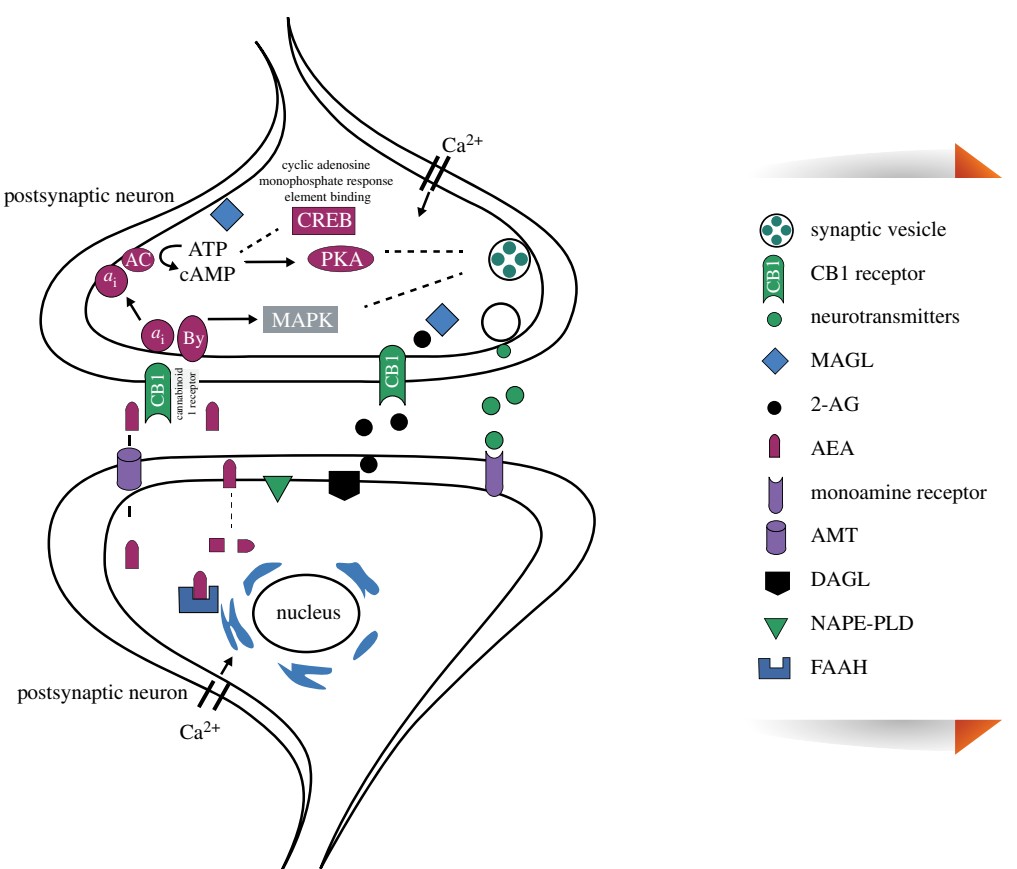

**Figure 1.** Schematic modeling of the endocannabinoid system. Adapted from Vinod & Hungund [23].

as retroactive signalling molecules to stimulate the presynaptic CB1 receptor. This leads to the activation of various effectors including adenylyl cyclase (AC), mitogen-activated protein kinase (MAPK), $K^+$ and $Ca^{2+}$ channels through $G\alpha i/o$ proteins. Inhibition of AC activity and decrease in cAMP content lead to a reduction in the activity of protein kinases (PKA) resulting in the modulation of the ion channels and neurotransmitter release. Selection mechanisms limit the activities of the anandamide and 2-AG, which is followed by hydrolysis with fatty acid amide hydrolase (FAAH) and monoacylglycerol lipase (MAGL), respectively [23]. The endocannabinoid system consists of endogenous cannabinoid receptor, ligands and proteins that are responsible for the metabolism and regulation of the endocannabinoid system. The first endocannabinoid system was characterized as AEA [24,25]. The second endocannabinoid system was discovered in 1994 [26–28]. 2-AG is present at a higher level in the mammalian central nervous system (CNS) compared with AEA. 2-AG and AEA act as full and partial agonists at the CB1 receptor (CB1R). The endocannabinoid system is abundantly present in the cerebral cortex, basal ganglia and limbic structures and exerts its effects mainly through CB receptors [25,29]. Some of this endocannabinoid system, especially AEA, also acts through the vanilloid receptors [24]. The endocannabinoid system consists of receptors and proteins that have the same pharmacologic action as tetrahydrocannabinol (THC), which is the main active constituent of cannabis. The cannabinoid downstream signalling is involved in various physiological functions that are a critical factor in depression and anxiety disorders, neuron deterioration and neurogenesis of the brain cells that are highly correlated with learning and memory function. Various lines of evidence strongly suggest that the endocannabinoid system anandamide and endocannabinoid signalling

cascades are regulated via cAMP response element (CRE) and the binding protein CREB mediated epigenetic alterations such as DNA methylation, histone hyperacetylation and deacetylation and miRNAs [25].

## 4. Epigenome and alcohol/ethanol

Exposure to alcohol at different ages can disturb chromatin function and impact neuron plasticity, but some of these effects may last longer [30]. The reports from several laboratories have highlighted the neurodegenerative effects of exposure to alcohol or alcohol and cannabinoid during adolescence and adulthood [31–33]. Pandey *et al*. [15] demonstrated that drinking during the adolescence period leads to a higher risk of developing alcohol addiction behaviour and other neuronal disorders in adulthood [34]. Nagre and his group studied the epigenetic events mediated by DNA methylation, which is a well-known epigenetic biomarker, that might lead to the major birth deformities that are induced by alcohol. Nevertheless, adolescence is an important developmental period in which there are changes that take place in the brain development process that impact neurotransmission, gene expression and synaptic remodelling, specifically the formation and pruning of axons, dendrites and synapses in various brain regions. Recent studies have shown the potential role of adolescent alcohol exposure-induced histone modifications and in stimulating alcohol intake during adulthood [32]. Another study found that knocking out the CB1 receptor protected mice against ethanol-induced impairment of DNA methylation [14]. Additionally, Sakharkar *et al*. [35] showed that intermittent ethanol exposure increased hippocampal [15,35] histone deacetylase (HDAC) which reduced the binding protein

CREB and histone H3K9 acetylation as well decreasing brain derived neurotropic protein (BDNF), a factor that maintains neuroplasticity and nerve growth factor. Crews *et al.* [30] described how the recent discovery of epigenetic mechanisms under environmental regulation may represent a significant portion of the genetic aspects of adolescent maturation. Recent studies have also indicated that alcohol can change gene expression through the epigenetic process in a way that is passed to offspring [32]. Animal studies demonstrated that exposure to alcohol changes epigenetics, endocrine–neuronal related genes and immune related genes expression for at least three generations [36]. Pandey *et al.* [15] demonstrated that adolescent intermittent ethanol (AIE) exposure leads to long-lasting increases in global HDAC activity, as well as specific increases in HDAC2 protein and mRNA, in the amygdala at adulthood [15]. It is associated with decreased H3K9 acetylation globally at the promoter regions of BDNF and ARC genes and markedly reduced dendritic spines [15]. The inhibitor trichostatin A (TSA) reversed the decrease in HDAC acetylation that was induced by AIE [32,35]. It was also reported that exposure to alcohol increased HDAC activity in the hippocampus and decreased CREB, which is an essential protein for healthy brain development; the downregulation of this protein triggers different behavioural disorders such as anxiety and depression [37,38]. Additionally, Nagre *et al.* [14] demonstrated that ethanol treatment to postnatal rodents on day 7 triggered the acetylation of H4 on lysine 8 (H4K8ace) at CB1R exon1, CB1R binding and CB1R-GTPγS binding. This report suggest the potential of epigenetic overlap between alcohol and cannabinergic activity.

# 5. Epigenetics and cannabis

The studies that have investigated cannabinoid-mediated epigenome modifications are limited. However, increasing cannabis legalization for both medical and recreational use has captured scientists' interest to reveal more epigenetic implications of this substance on different systems during various life stages. Earlier studies showed that THC treatment significantly altered histone methylation and acetylation. One study analysed the effect of THC exposure on histone biomarkers such as H3K36me3, H3K9me3 and H3K9ac in several immunogenic phase analyses. Interestingly, these biomarkers impacted expression of certain gene promoters, suggesting further downstream gene alteration. They reported that THC exposure reduced significantly expression of Brca2, a tumour suppressor gene, Rorc, Tbx-21, Ifn-γ and IL-2 promoters, while it increased IL-4, IL-5 and CBX-1 promoter expression via the mentioned histone methylated regions, using staphylococcal enterotoxin B induced T-cell activation. Although, the global histone methylation was not altered by THC, the study showed that THC may modulate immune response through epigenetic regulation involving histone modifications [39]. Another study showed the dysregulation of striatal mRNA levels in adolescents and adult offspring with parental THC exposure [40]. They analysed 1027 differentially methylated regions (DMRs) at various CpGs that are located within gene introns, exons and intervals in the first offspring with parental THC exposure. The analysis showed downregulation of gene promoters, and also showed alteration of mRNA expression linked to DMR-associated genes involved in glutamatergic synaptic regulation within the nucleus

accumbens [40]. Another study looked at cannabis effect on infants' DNA methylation in the dopamine receptor promoter at CpG units. There was no significant evidence that cannabis impacted the methylation in these specific areas [41].

## 5.1. Epigenetics and neuronal development with alcohol/ethanol and cannabis induced neuro-modification in adolescence and adulthood

Emerging evidence is demonstrating that the endocannabinoid system plays a major role in substance use, including alcohol use [42]. Acute ethanol exposure has been shown to upregulate CB1R, while chronic exposure downregulated CB1R in other studies [14,31,42]. Emerging studies have demonstrated the impact of cannabis, alcohol and other abused drugs on the inherited epigenome in further generations [40,43]. For example, exposure to environmental insults altered DNA methylation persistently in further generations [40,43–46]. Despite the lack of solid understanding about the mechanistic effects of the involvement of the cannabinoid system and alcohol use to cross generations, there are some studies that have revealed novel pathways. Several studies correlated epigenome cascades through DNA methylation and histone acetylation that were altered by ethanol exposure in the endocannabinoid system [14,30–32]. It was demonstrated that adolescent exposure to THC in rat models affected the reward signalling pathway and resulted in an abnormal behaviour pattern because of the epigenetic dysregulation that had crossed to the first offspring who had no direct exposure to any substance [13]. Nagre *et al.* [14] and his group demonstrated that ethanol exposure reduced DNA methyltransferases (DNMT1 and DNMT3A) by activation of caspase 3, an enzyme that is involved in the inflammatory process. This observation was not seen in CB1 knock out mice. On the other hand, a longitudinal pregnancy cohort of 1634 families studied effects of perinatal maternal gestational cannabis on infants' health through investigating offspring DNA methylation. The study focused on the offspring dopamine receptor gene DRD4 promoter in infants, and found that there is no potential inherited risk in the infants [41]. The study suggested that investigating a larger sample and other genes might have different results [41]. Other studies found the persistence of adolescent-like responses to alcohol in adulthood, increased adult anxiety and increased adult alcohol consumption through epigenetic signalling [15,32,47]. The studies suggested that massive binge drinking in adolescence has long-lasting effects on the adult brain and behaviour that is induced by epigenetic modifications [15,30,35,43,47]. This lasting effect is induced by altering histone acetylation of the CB1 receptor that also triggers further downstream signalling effects [32]. Moreover, Crews and his group correlated binge drinking inducing neuronal disorders with epigenetic modifications and their effect later in life [30]. Sakharkar *et al.* [35] suggested that various epigenetic pathways triggered ethanol-induced functional changes of the brain in the adolescent developmental period. Some of these epigenetic processes are likely to be involved in certain behavioural and synaptic alterations shown during adolescence. Figure 2 shows the proposed mechanism of epigenetic reprogramming by alcohol exposure at this early stage of life. Alcohol-induced adolescence modulations are mediated by epigenetic reprogramming that cause dysregulated neuronal function such as axonal outgrowth, dendrite

royalsocietypublishing.org/journal/rsob Open Biol. 9: 180115

royalsocietypublishing.org/journal/rsob   *Open Biol.* **9**: 180115

## NORMAL EPIGENETIC PROGRAMMING

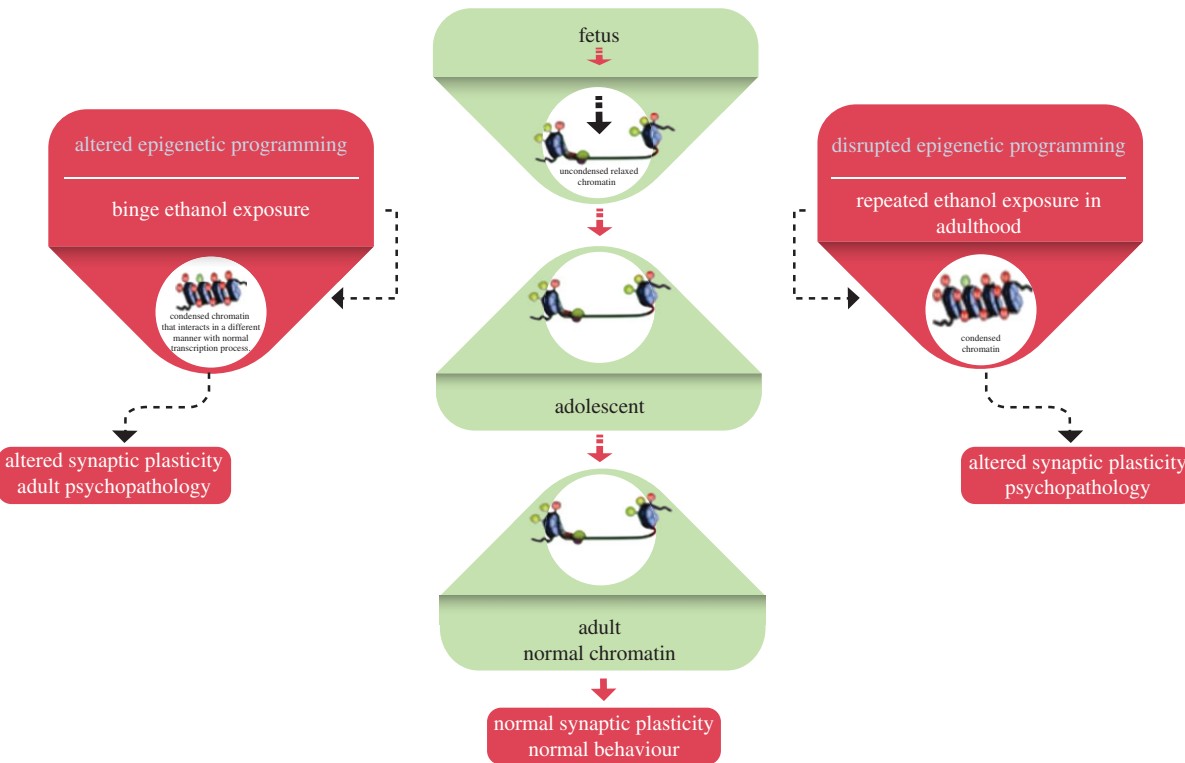

**Figure 2.** Mechanism of epigenetic reprogramming caused by alcohol exposure. From Kyzar & Pandey [32].

formation and neuronal maturation, which then impact the behavioral process and synaptic plasticity that eventually regulates normal behaviours [32,36]. Previous evidence shows that exposure to alcohol in this early stage potentially remodels chromatin constitution and predisposes individuals to neuronal disorders such as anxiety and depression. Exposure to ethanol increases the activity of the endocannabinoid system in the brain, leading to decreased CB1 binding and uncoupling that is directly associated with consumption of ethanol.

### 5.2. Epigenetic biomarkers

To meet the difficulties we face in identifying substance use disorders, it is necessary to identify key epigenetics biomarkers. A biomarker is a measurable molecule that facilitates identifying the abnormal biological process in comparison to normal physiology, pathophysiology or disease. Ideally the biomarker should possess high sensitivity and specificity. In the domain of psychiatry and substance use disorders, new biomarkers that improve on existing technology are in the development process. Epigenetic biomarkers include DNA methylation at CpG islands, histone tail modifications, small non-coding RNAs and open versus closed chromatin packing. The basic concepts of epigenetics have been reviewed previously [48–50] and will not be discussed in detail here. Technically, epigenetic remodelling affects the regulation of the genes, which in turn may affect cellular structure and function. Clinical and preclinical studies have shown that changes in the expression of genes such as BDNF alter the characteristics of use of drugs such as alcohol, and cannabis [26,51]. Therefore, epigenetic biomarkers indicate changes in gene expression that contribute to our understanding of addictions. DNA methylation is one of the epigenetic biomarkers that are

most probably linked to development of substance use disorder. The methylation process is one that demonstrates mental stability. DNA methylation is a key component in multiple gene regulation processes (figure 3), via several mechanisms that involve transcription and splicing factors [27,43]. DNA methylation regulation was studied in different behavioural and physiological phenotypes in animal models, with relevance to behavioural and physiological alterations previously described in our model [40]. Additionally, many studies have revealed that DNA methylation is a major process in normal neuronal and brain development [28,43,52]. Moreover, DNA methylation dysregulation has been observed in addiction, anxiety, depression, autism, schizophrenia and bipolar disorder [3,52–54]. DNA methylation assays can be used in different tissues nowadays as it is more accurate and affordable than before. In contrast, assays of histone modifications and RNAs are expensive and time-consuming. Changes in global DNA methylation, often measured via digestion and analysis of repetitive DNA motifs distributed throughout the genome such as long interspersed element-1 (LINE-1), have been reported in cancer, but these techniques have poor specificity regarding substance abuse [55]. Methylation at a given CpG residue is mostly reported as a percentage of methylation [55]. For application to brain disorders, however, brain tissue is not obtainable in a clinical setting [55]. Nagre *et al*. [14] studied the effect of alcohol on epigenetic cascades in animals that showed disturbance of different markers when exposed to alcohol in the postnatal period. They demonstrated novel epigenetic modification in the fetal alcohol spectrum disorder mouse model. They exposed mice to alcohol on day 7, which induced activation of caspase 3. Caspase 3 is a marker that triggers the downstream inflammation cell signalling process. Nagre *et al*.

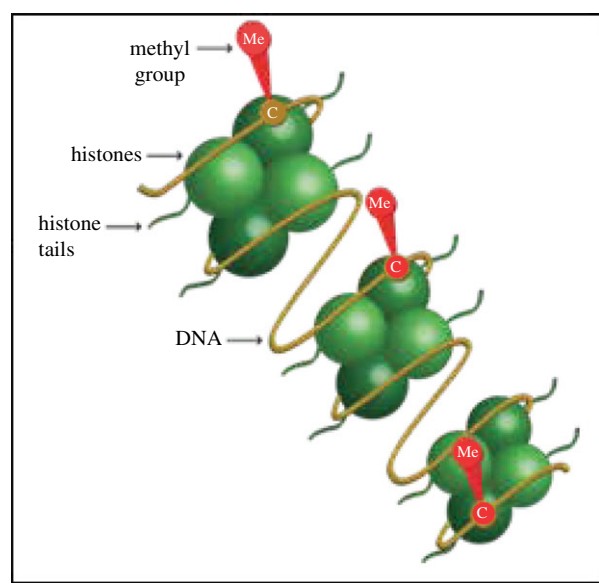

**Figure 3.** DNA methylation (www.epigentek.com/catalog/dna-methylation-antibodies-c-35_104_26.html).

**Figure 4.** DNA hydroxymethylation.

found that alcohol treatment impaired DNA methylation through reduction of different epigenetic markers such as DNA methyltransferase (DNMT1 and DNMT3A) levels; besides, they reported that inhibiting caspase 3 activity before exposing mice to alcohol treatment rescued DNMT1 and DNMT3A proteins as well as DNA methylation levels. Moreover, they demonstrated that inhibition of histone methyltransferase (G9a) action and blocking CB1R, before alcohol exposure, respectively, inhibited caspase 3, which in turn rescued the DNMT1 and DNMT3A proteins and DNA methylation. There was no significant reduction of DNMT1 and DNMT3A proteins and DNA methylation in CB1R null mice. Nagre *et al*. showed that ethanol-induced activation of caspase 3 impairs DNA methylation through DNMT1 and DNMT3A in the neonatal mouse brain, and such impairments are absent in CB1R null mice [14]. Another biomarker is histone acetylation, which has been widely studied in cancer aetiology through different mechanisms, one of them being the transcriptional control and regulation of checkpoints by HDAC1 and HDAC2 [56]. Histones can restructure chromatin by several mechanisms such as phosphorylation, ubiquitination, acetylation and methylation [1,57,58]. In addition, histone code combinations may alter the transcriptional process at many levels and in many ways in combinations of different factors and sites that add up to the complexity of histone acetylation and deacetylation [57]. One of the recognized histone mediated process is the interaction with the amine group at the lysine site in H3 AND H4 that most probably is associated with fostering the transcription process [57]. H3K4 is one of the histone binding sites that is linked to enhancement of the transcription process in the neuronal system. On the other hand, H3K9 is a repressive marker that is linked to downregulation of transcription at this binding site.

## 5.3. Epigenetic methods

### 5.3.1. DNA methylation

DNA methylation is a major epigenetic modification involving the addition of a methyl group to the C5 position of cytosine at CpG (cytosine guanine) islands [5,10] by DNA methyltransferase (DNMT) to form 5-methylcytosine (5-mC). 5-mC is an important epigenetic marker that enables expression of some genes and represses others. Studies have shown that it could be passed on to the next generation through cell division. Techniques to quantify 5-mC have been developed over the years and can vary from heat-based to chemical-based methods (www.epigentek.com/catalog/dna-methylation-antibodies-c-35_104_26.html). Additional methods for investigating DNA methylation include an antibody-based technique known as methylated DNA immunoprecipitation (MeDIP) that is used for studying gene-specific DNA methylation on a large genomic scale, and activation/inhibition assays that quantify DNMT, used for studying gene or sequence-specific DNA methylation (www.epigentek.com/catalog/dna-methylation-antibodies-c-35_104_26.html).

DNA methylation was one of the first discovered epigenetic markers; it plays an essential role in human development and birth abnormalities. Additionally, DNA hydroxymethylation (figure 4), that is caused by oxidation of 5-mC group through the ten eleven translocation (TET) enzyme is involved in controlling gene expression as well as DNA demethylation. Earlier in this review, we gave a brief overview of DNA methylation in the biomarker section and highlighted how important it is in epigenetic studies. It is key to vital biological processes, and it crosses functional chromatin elements.

### 5.3.2. Chromatin immunoprecipitation assay

Chromatin immunoprecipitation (ChIP) assay is one of the most commonly used epigenetic assays, and studies the protein–DNA environmental interaction with the chromatin content. This assay can be used to identify the different proteins that are located in different regions [59,60]. ChIP assay can be used to detect the arrangement of the protein relative to specific promoters or factors or to quantify the amount of associated histone around the gene [61,62]. Also, ChIP assay is used to study transcriptional factors, DNA repair protein and the DNA replication process. Basic PCR technique is used to identify the DNA sequences or a specific region of the gene that is associated with a protein or histone modification [59,60].

The ChIP technique and different types of sequencing which are empowered by DNA and RNA sequencing of chromatin will motivate further studies of effects of drug treatment on gene expression. High-throughput genomic studies of chromatin modification will lead to more understanding of how drugs predispose an individual to epigenetic modification and gene expression imbalance [3,5]. Figure 5 shows the ChIP assay process.

### 5.3.3. Histone deacetylase assay

'Histone deacetylases (HDACs), also known as lysine deacetylases, are a family of enzymes that remove acetyl groups

royalsocietypublishing.org/journal/rsob Open Biol. **9**: 180115

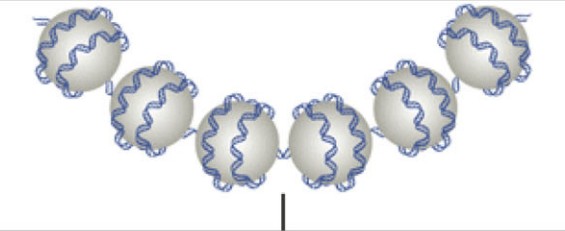

cells are fixed with formaldehyde to cross-link histone and non-histone proteins to DNA.

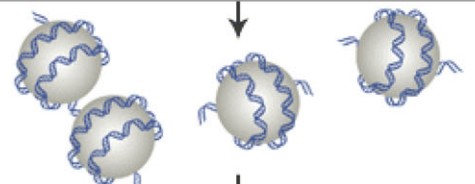

chromatin is digested with micrococcal nuclease into 150–900 bp DNA/protein fragments.

antibodies specific to histone or non-histone proteins are added and the complex co-precipitates and is captured by protein G agarose or protein G magnetic beads.

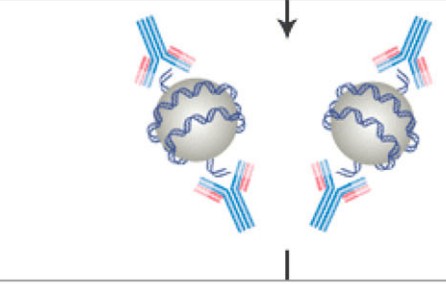

cross-links are reversed, and DNA is purified and ready for analysis.

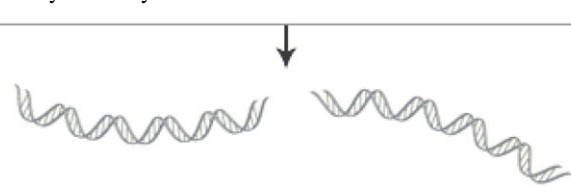

**Figure 5.** Chromatin immunoprecipitation assay process (www.cellsignal.com/contents/resources-applications-chromatin-immunoprecipitation/overview-of-chromatin-ip-assay-methodology/chip-assay-overview).

from histones such as H2A, H2B, H3 and H4; thus, they are involved in regulating gene expression.

These epigenetic enzymes are highly linked with chromatin structure. Removal of acetyl groups by HDACs tightens the chromatin structure, which will limit the transcription and recruitment of further molecules. The DNA that surrounds the histones is more compact, which can reduce the transcription of essential genes. HDACs are also tightly involved in other cell cycle signalling mechanisms and cell differentiation and growth, thus it is often seen as a cancer biomarker. Moreover, in the development of human cancer. By studying the activity and inhibition of HDACs, researchers can gain a better understanding of the impact of histone deacetylation on diseases and various cellular processes (www.epigentek.com/catalog/dna-methylation-antibodies-c-35_104_26.html).

# 6. Epigenetic analysis

## 6.1. Histone acetylation quantification

Acetylation of the histones including H3 and H4 by histone acetyltransferase (HAT) will add an acetyl group to the histone structure. Studies have shown that open or loosen chromatin will enable the transcriptional machinery and thus increase transcription and gene expression.

## 6.2. Histone methylation quantification

Methylation of the histone by adding a methyl group can turn some genes on and off, thus it is a very useful assay in gene expression studies.

## 6.3. Future direction

The study of epigenetic mechanisms is a new direction in understanding what shapes our phenotypes and characteristics and the hereditary hypothesis. The nature of the lasting effect that alters neuron function following chronic exposure to abused drugs is still not established. There are too many theories that we are still learning about, such as, how long it takes after drug exposure to trigger modifications in chromatin remodelling. Many of the genes that display different regulation in response to drug exposure show a temporary change and then resumes their normal process after a certain period. Thus, it is essential to take the lasting effect on gene expression into consideration in the theory of epigenetics being responsible for turning some genes on and other genes off. Furthermore, investigation is needed to identify which genes show effects mediated by epigenetic and chromatin remodelling that appear in the next offspring. Which regions of altered genes are impacted by drug-induced changes in the chromatin structure and how these chromatin modifications impact our genotypes in response to certain treatment exposure [5] requires further study, as well as how chromatin modifications impact cell differentiation in response to drug exposure. Further, understanding of the combined effects of alcohol and cannabis exposure together and whether they have an additive epigenetic effect or one increases the effect of the other, in which epigenetic area, and which genes are involved will enable exploration of targets for future therapeutic intervention for addiction.

# 7. Conclusion

In summary, the increase in adolescent and adult exposure to alcohol and cannabis is an increasing worldwide problem, but the outcome of that exposure and co-exposure on brain functions are poorly understood. This has raised the need to study the long-term impact of the combined exposure to alcohol and cannabis among adolescents and adults, given the prevalence of their abuse, and the potential harmful lasting effect needs to be understood and addressed [13]. We have shown that epigenetics is a new area of science for study of substance use disorders that is still at a preliminary stage for alcohol, and for cannabis fewer studies have been done. However, substance use disorders continue to cause significant consequences on a worldwide basis, and thus more study is urgently needed to provide clinicians and

researchers with alternatives to detect and treat these disorders [3,5]. Such investigations might help in the development of novel therapeutics for future treatments of addiction. Finally, the work on epigenetic studies has expanded our understanding by documenting distinct regulatory patterns of chromatin modifications and the associated transcription machinery associated with downstream signalling pathways in response to different stimuli. However, many important questions remain unanswered regarding the extent of epigenetic modifications and whether they are inherited, resulting in a lasting effect on future generations. Additionally, the impact on development of behavioural characteristics enables drug addiction during different life stages [3,5]. We have more to understand regarding the molecular mechanisms that are linked with chromatin remodelling as a critical regulator that has been associated with substance use induced transcriptional and behavioural changes. The application of the epigenetic theory to reverse the normal cellular process is evolving with more theories to be established regarding chromatin regulation and gene expression.

Data accessibility. This article has no additional data.
Authors' contributions. Y.E.D. and M.M.A. wrote, reviewed and approved the version of this manuscript.
Competing interests. The authors declare that they have no competing interests.
Funding. The review was supported by Zewail City of Science and Technology University and North Carolina Central University. The funding bodies do not have a role in the design of the study, collection, analysis and interpretation of data or in writing the manuscript.

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
