## [Reviewer comments · Open Biology]

Review History

RSOB-18-0115.R0 (Original submission)

Review form: Reviewer 1

Recommendation

Major revision is needed (please make suggestions in comments)

Are each of the following suitable for general readers?

- a) **Title**
Yes
- b) **Summary**
Yes
- c) **Introduction**
Yes

Is the length of the paper justified?

No

Should the paper be seen by a specialist statistical reviewer?

No

Is it clear how to make all supporting data available?

Not Applicable

Is the supplementary material necessary; and if so is it adequate and clear?

Yes

Do you have any ethical concerns with this paper?

No

Comments to the Author

Good start but the flow of the manuscript for a review article still needs more work. I found a lot of typos in the manuscript in the first couple of pages. In addition, I would like the tables and images to be original or reformatted by the author. It seems like the authors just copied and pasted the material, I am not sure if that material can be reprinted without special permission from the referenced authors. The flow of the manuscript needs additional work. The images and tables are truncated at times. The section titles and font not consistent through the manuscript. Major revisions needed.

Review form: Reviewer 2

Recommendation

Major revision is needed (please make suggestions in comments)

Are each of the following suitable for general readers?

- a) **Title**
Yes
- b) **Summary**
Yes
- c) **Introduction**
Yes

Is the length of the paper justified?

Yes

Should the paper be seen by a specialist statistical reviewer?

No

Is it clear how to make all supporting data available?

Not Applicable

Is the supplementary material necessary; and if so is it adequate and clear?

Not Applicable

Do you have any ethical concerns with this paper?

No

Comments to the Author

This is an interesting and useful for readers to understand epigenetics mechanism, and the effects of ethanol and cannabinoid. The review is fair and covers most of the key papers. I have a few suggestions, which the authors are free to consider.

The comments are listed below:

1. The review title: The current review has discoursed on the Epigenetic modulation of ethanol and cannabinoid separately; however, the review has not discoursed the Epigenetic modulation of ethanol and cannabinoid co-exposure as expected.
2. In "Introduction", it will be better for readers to add more references to determine why it is interesting and important to review "Epigenetic modulation of ethanol and cannabinoid co-exposure"
3. The subtitle "Epigenetic and neuronal development to Alcohol and Cannabis induced neuro modification in Adolescence and Adulthood" only discoursed the effect of ethanol, and do not have any Cannabis' content. It is good to merge this part into the subtitle "Epigenome and Alcohol" as "Epigenetic and Alcohol/Ethanol"
4. In the "Epigenetic and Cannabis" part, there is only one reference paper (number 25); more papers should be added to determine the epigenetic mechanism of cannabinoid. In addition, Reference number 25, which is worry author's order in the reference.
5. Currently, it is very unclear for understanding the interaction and epigenetic mechanisms of alcohol and cannabinoid exposure. If authors want to further discuss the interaction between ethanol and cannabinoid, possibly need more references in the review, such as Parira T et al., "Epigenetic Interactions between Alcohol and Cannabinergic Effects: Focus on Histone Modification and DNA Methylation", 2017...
6. Hope that authors can re-think the review structure. Suggesting for the order of the subtitle" 1. Introduction; 2. Epigenetics definition and mechanism; 3. Epigenetics modulation of ethanol and cannabinoid exposure, 3.1 Epigenetics and ethanol, 3.2 Epigenetics and cannabinoid; 3.3 Interaction between alcohol and cannabinoid or co-exposure; 4. Epigenetics analysis, 4.1 Epigenetics biomarkers, 4.2 Epigenetics methods; 5. Conclusion
7. References: It is the best way of sorting the sources numerically, and by the appearing sequence.

Some references are hard to figure them out in the reference, such as:

- a. Page 3, Crews et al., Sakharakar et al., and Pandey et al..
- b. Page 7, Hurd et al. and Szutorisz et al.(2015)
- c. Page 9, Craiu 2013;Spear 2013
- d. Page 15, Ann N Y Acad Sci..... Page 11?

e. And more....

Decision letter (RSOB-18-0115.R0)

03-Aug-2018

Dear Dr Dobs,

We are writing to inform you that the Editor has reached a decision on your manuscript RSOB-18-0115 entitled "The Epigenetic modulation of Ethanol and Cannabinoid co-exposure during the Adolescent", submitted to Open Biology.

As you will see from the reviewers' comments below, there are a number of criticisms that prevent us from accepting your manuscript at this stage. The reviewers suggest, however, that a revised version could be acceptable, if you are able to address their concerns. If you think that you can deal satisfactorily with the reviewer's suggestions, we would be pleased to consider a revised manuscript.

The revision will be re-reviewed, where possible, by the original referees. As such, please submit the revised version of your manuscript within six weeks. If you do not think you will be able to meet this date please let us know immediately.

When submitting your revised manuscript, please respond to the comments made by the referee(s) and upload a file "Response to Referees" in "Section 6 - File Upload". You can use this to document any changes you make to the original manuscript. In order to expedite the processing of the revised manuscript, please be as specific as possible in your response to the referee(s).

Please see our detailed instructions for revision requirements
<https://royalsociety.org/journals/authors/author-guidelines/>

Sincerely,

The Open Biology Team
mailto: openbiology@royalsociety.org

Reviewer(s)' Comments to Author(s):

Referee: 1

Comments to the Author(s)

Good start but the flow of the manuscript for a review article still needs more work. I found a lot of typos in the manuscript in the first couple of pages. In addition, I would like the tables and images to be original or reformatted by the author. It seems like the authors just copied and pasted the material, I am not sure if that material can be reprinted without special permission from the referenced authors. The flow of the manuscript needs additional work. The images and tables are truncated at times. The section titles and font not consistent through the manuscript. Major revisions needed.

Referee: 2

Comments to the Author(s)

This is an interesting and useful for readers to understand epigenetics mechanism, and the effects of ethanol and cannabinoid. The review is fair and covers most of the key papers. I have a few suggestions, which the authors are free to consider.

The comments are listed below:

1. The review title: The current review has discoursed on the Epigenetic modulation of ethanol and cannabinoid separately; however, the review has not discoursed the Epigenetic modulation of ethanol and cannabinoid co-exposure as expected.
2. In "Introduction", it will be better for readers to add more references to determine why it is interesting and important to review "Epigenetic modulation of ethanol and cannabinoid co-exposure"
3. The subtitle "Epigenetic and neuronal development to Alcohol and Cannabis induced neuro modification in Adolescence and Adulthood" only discoursed the effect of ethanol, and do not have any Cannabis' content. It is good to merge this part into the subtitle "Epigenome and Alcohol" as "Epigenetic and Alcohol/Ethanol"
4. In the "Epigenetic and Cannabis" part, there is only one reference paper (number 25); more papers should be added to determine the epigenetic mechanism of cannabinoid. In addition, Reference number 25, which is worry author's order in the reference.
5. Currently, it is very unclear for understanding the interaction and epigenetic mechanisms of alcohol and cannabinoid exposure. If authors want to further discuss the interaction between ethanol and cannabinoid, possibly need more references in the review, such as Parira T et al., "Epigenetic Interactions between Alcohol and Cannabinergic Effects: Focus on Histone Modification and DNA Methylation", 2017...
6. Hope that authors can re-think the review structure. Suggesting for the order of the subtitle" 1. Introduction; 2. Epigenetics definition and mechanism; 3. Epigenetics modulation of ethanol and cannabinoid exposure, 3.1 Epigenetics and ethanol, 3.2 Epigenetics and cannabinoid; 3.3 Interaction between alcohol and cannabinoid or co-exposure; 4. Epigenetics analysis, 4.1 Epigenetics biomarkers, 4.2 Epigenetics methods; 5. Conclusion

7. References: It is the best way of sorting the sources numerically, and by the appearing sequence.

Some references are hard to figure them out in the reference, such as:

- a. Page 3, Crews et al., Sakharakar et al., and Pandey et al..
- b. Page 7, Hurd et al. and Szutorisz et al.(2015)
- c. Page 9, Craiu 2013;Spear 2013
- d. Page 15, Ann N Y Acad Sci..... Page 11?
- e. And more....

Author's Response to Decision Letter for (RSOB-18-0115.R0)

See Appendix A.

RSOB-18-0115.R1 (Revision)

Review form: Reviewer 1

Recommendation

Accept as is

Are each of the following suitable for general readers?

- a) **Title**
Yes
- b) **Summary**
Yes
- c) **Introduction**
Yes

Is the length of the paper justified?

Yes

Should the paper be seen by a specialist statistical reviewer?

Yes

Is it clear how to make all supporting data available?

Yes

Is the supplementary material necessary; and if so is it adequate and clear?

Yes

Do you have any ethical concerns with this paper?

No

Comments to the Author

Thank you for making necessary changes.

Decision letter (RSOB-18-0115.R1)

11-Dec-2018

Dear Dr Dobs

We are pleased to inform you that your manuscript RSOB-18-0115.R1 entitled "The Epigenetic modulation of Alcohol/ Ethanol and Cannabis Exposure/Co- exposure During Different Stages." has been accepted by the Editor for publication in Open Biology. The reviewer(s) have recommended publication, but also suggest some minor revisions to your manuscript. Therefore, we invite you to respond to the reviewer(s)' comments and revise your manuscript.

Please submit the revised version of your manuscript within 14 days. If you do not think you will be able to meet this date please let us know immediately and we can extend this deadline for you.

- 1) A text file of the manuscript (doc, txt, rtf or tex), including the references, tables (including captions) and figure captions. Please remove any tracked changes from the text before submission. PDF files are not an accepted format for the "Main Document".
- 2) A separate electronic file of each figure (tiff, EPS or print-quality PDF preferred). The format should be produced directly from original creation package, or original software format. Please note that PowerPoint files are not accepted.
- 3) Electronic supplementary material: this should be contained in a separate file from the main text and meet our ESM criteria (see <http://royalsocietypublishing.org/instructions->

authors#question5). All supplementary materials accompanying an accepted article will be treated as in their final form. They will be published alongside the paper on the journal website and posted on the online figshare repository. Files on figshare will be made available approximately one week before the accompanying article so that the supplementary material can be attributed a unique DOI.

Online supplementary material will also carry the title and description provided during submission, so please ensure these are accurate and informative. Note that the Royal Society will not edit or typeset supplementary material and it will be hosted as provided. Please ensure that the supplementary material includes the paper details (authors, title, journal name, article DOI). Your article DOI will be 10.1098/rsob.2016[last 4 digits of e.g. 10.1098/rsob.20160049].

4) A media summary: a short non-technical summary (up to 100 words) of the key findings/importance of your manuscript. Please try to write in simple English, avoid jargon, explain the importance of the topic, outline the main implications and describe why this topic is newsworthy.

Images

Data-Sharing

It is a condition of publication that data supporting your paper are made available. Data should be made available either in the electronic supplementary material or through an appropriate repository. Details of how to access data should be included in your paper. Please see <http://royalsocietypublishing.org/site/authors/policy.xhtml#question6> for more details.

Data accessibility section

Sincerely,

The Open Biology Team
<mailto:openbiology@royalsociety.org>

ditage Insights by clicking on the following link: <https://www.surveymonkey.com/r/author-perspectives-on-academic-publishing-royal-society>

This should take no more than 15 minutes and you will have the opportunity to enter a prize draw. We hope these results will provide us with valuable insights we can use to improve our service.

Reviewer(s)' Comments to Author:

Referee: 1

Comments to the Author(s)

Thank you for making necessary changes.

Author's Response to Decision Letter for (RSOB-18-0115.R1)

Thank you for reviewing , accepting my Manuscript and suggesting further amendments. I have developed what I have been asked to submit.

1- The Text Files

2- A separate PDF high resolution for the figures.

3- I do not have supplementary figures or files that is attached in my Manuscript.

4- I developed a media Summary less than a 100 words in a simplified way for readers.

Once again thanks for my Reviewers for their time and for Open Biology for providing a platform for open access where scientists and readers have an access to valuable articles and knowledge.

Thanks for Royal Society for empowering authors and science to unleash their potential in this great place.

Sincerely,

Yasmina Dobs

Decision letter (RSOB-18-0115.R2)

04-Jan-2019

Dear Dr Dobs

We are pleased to inform you that your manuscript entitled "The Epigenetic Modulation of Alcohol/ Ethanol and Cannabis Exposure/ Co- exposure During Different Stages." has been accepted by the Editor for publication in Open Biology.

Article processing charge

Please note that the article processing charge is immediately payable. A separate email will be sent out shortly to confirm the charge due. The preferred payment method is by credit card; however, other payment options are available.

Sincerely,

The Open Biology Team
mailto: openbiology@royalsociety.org

ditage Insights by clicking on the following link: <https://www.surveymonkey.com/r/author-perspectives-on-academic-publishing-royal-society>

This should take no more than 15 minutes and you will have the opportunity to enter a prize draw. We hope these results will provide us with valuable insights we can use to improve our service.

Appendix A

For Reviewer 1:

Thanks for addressing this. I agree that there were typos and some grammatical errors, I proof read the Manuscript several times with Native English speaker to make sure that the new version meets the publication standards and readers satisfaction. I also re illustrated figure 2 and obtained copy-write approvals for all the illustrations and figures from original authors and the publishers. I also changed the formatting as per the journal standards and I reformatted all citations as per the Journal style and as per the appearance order in my Manuscript.

For Reviewer 2:

1.The review title: The current review has discoursed on the Epigenetic modulation of ethanol and cannabinoid separately; however, the review has not discoursed the Epigenetic modulation of ethanol and cannabinoid co-exposure as expected

My response:

I changed the Manuscript title to " The Epigenetic modulation of Alcohol/ Ethanol and Cannabis Exposure/Co- exposure During Different Stages." and I expanded the discussion on the co exposure area. However, there are not many paper out there that discuss the Co exposure of Ethanol and Cannabis together this is why the point of the Manuscript is to bring attention to be researched.

2.In “Introduction”, it will be better for readers to add more references

My response

I agree, I added around 10 more references to the review.

3- The subtitle “Epigenetic and neuronal development to Alcohol and Cannabis induced neuro modification in Adolescence and ...

My response

I addressed that in the subtitle and in the discussion under that section. I added more Cannabis discussion and references, however there are few studies that are discussing the Cannabis effect within Epigenome area. I added What I believed that its relevant to the discussion.

4.In the “Epigenetic and Cannabis” part, there is only one reference paper (number 25)

My response

I added reference (43,44,45) in addition to 25. I also reordered the references as per the appearance

5-. If authors want to further discuss the interaction between ethanol and cannabinoid, possibly need more references in the review, such as Parira T et al., “Epigenetic Interactions between Alcohol.

My Response

Thanks for providing this study,

I included it in my discussion. I also found that Parira et al review cited similar articles such as Nagre et al and Subbanna et al when the author discussed the Epigenetic part.

6-Hope that authors can re-think the review structure. Suggesting for the order of the subtitle” 1. Introduction; 2. Epigenetics definition and mechanism; 3. Epigenetics modulation of ethanol and cannabinoid exposure, 3.1 Epigenetics and ..

My response

I reordered it this way. Thanks for highlighting it. I believe that it does make more sense this way.

7.References: It is the best way of sorting the sources numerically, and by the appearing sequence

My Response,

I listed the references numerically and I used Vancouver style according to the journal’s guidelines.